**Data Availability Statement:** All relevant data are within the paper and its Supporting information files.

**Funding:** The authors received no specific funding for this work.

# Treatment outcome of patients with prosthetic stuck valves at the Cardiac Center of Ethiopia

**Kefelegn Dejene Tadesse[1], Meron Gebru[2], Atnafu Mekonnen Tekleab[3]\***

**1** Consultant Interventional Cardiologist, Cardiac Center of Ethiopia, Addis Ababa, Ethiopia, **2** General Practitioner, Cardiac Center of Ethiopia, Addis Ababa, Ethiopia, **3** Department of Pediatrics and Child Health, St Paul's Hospital Millennium Medical College, Addis Ababa, Ethiopia

* atnemekonnen@yahoo.com

## Abstract

### Background

Prosthetic Valve Thrombosis (PVT) is rare but life threatening condition which requires urgent intervention. Patient treatment outcome is not well studied in resource limited settings and the current study aims to explore the treatment outcome of patients with PVT at the Cardiac Center of Ethiopia.

### Methods

The study was conducted at the Cardiac Center of Ethiopia which provides heart valve surgery. All patients who were diagnosed and managed for PVT in the center during the period July 2017 to March 2022 were included in the study. Data were collected through chart abstraction by using a structured questionnaire. Data analysis was done using SPSS version 20.0 for windows software.

### Result

Eleven patients (13 episodes of stuck valve) with PVT were included in the study and nine of them were female. The median age was 28 years old (IQR 22.5–34.0) with the youngest and oldest patients being 18 and 46 years old respectively. All the patients had bi-leaflet prosthetic mechanical valves (10 at mitral valve, two at aortic and mitral and one at aortic positions). The median duration of valve replacement before having PVT was 36 months (IQR 5–72). All patients reported good adherence to anticoagulant therapy; yet only five had optimal INR value. Nine patients presented with failure symptoms. Eleven patients received thrombolytic therapy and nine of them responded to it. One patient operated for failed thrombolytic therapy. Two patients responded to heparinization and optimization of anticoagulant therapy. Of the ten patients who received streptokinase, two of them developed fever and one patient developed bleeding as a complication of the treatment. All the patients survived hospital discharge.

**Competing interests:** I have read the journal's policy and one of the authors of this manuscript have the following competing interest: Dr Atnafu Mekonnen Tekleab is serving as Academic Editor of PLOS ONE Journal. However, this does not alter our adherence to PLOS ONE policies on sharing data and materials.

## Conclusion

Prosthetic valve thrombosis was accompanied by sub-optimal anticoagulant therapy. Most patients responded to medical therapy alone.

## Introduction

Rheumatic valvular pathology is very common in developing countries and often require surgical replacement with prosthetic valves [1]. Although Prosthetic valve thrombosis (PVT) is rare (seen in 0.3–1.3% of prosthetic valve in developed countries), it is associated with high mortality and morbidity [2]. Prosthetic valve type, anticoagulation status, valve position, the presence of prothrombotic states such as pregnancy, atrial fibrillation, and/or ventricular dysfunction determine the risk of PVT. However, inadequate anticoagulant therapy is the leading cause of PVT [3].

Early diagnosis and appropriate treatment is very vital [4]. Although surgical approach, fibrinolysis, and heparin treatment are considered as treatments of choice, the clinical status of the patient and the valvular condition determine the option of treatment [2].

Fibrinolytic agents are among the treatment options [5] though they are associated with a high risk of emboli, bleeding, and high mortality rates [6–8]. Complications related to use of fibrinolytic therapy can be decreased with a low dose and slow infusion of fibrinolytic therapy [9]. In 1971, Luluaga et al were the first to use thrombolytic therapy in PVT and they used Streptokinase to treat thrombosis of the tricuspid valve prosthesis. Later, Baille et al reported the use of a thrombolytic agent in a patient with aortic PVT [10].

Prosthetic valve replacement surgery is one of the open heart surgeries being offered at the Cardiac Center of Ethiopia. Following the valve replacement surgery, achieving optimal anticoagulation status during patient follow up is a challenge for several reasons. Sometimes patients present with emergency condition due to prosthetic stuck valve which is a life threatening condition in the absence of urgent intervention. There are limited studies particularly in developing countries regarding the clinical profile and treatment outcomes of patients with prosthetic stuck valves. This study aimed to explore the treatment outcome of patients who had prosthetic stuck valve and who were treated at the Cardiac Center of Ethiopia, Addis Ababa, Ethiopia.

## Methods

### Study setting

The study was conducted at the Cardiac Center of Ethiopia (CCE), found in Addis Ababa, Ethiopia. CCE is a non-profit charity organization located at the heart of Addis Ababa. It was established in 2009 and since then the center has performed hundreds of valve replacement surgeries.

### Study design and study period

Data were collected through chart abstraction during the month of May 2022. A retrospective institutional based cross-sectional study design was used and patients who developed stuck valve and who were treated in the center between the periods July 2017 to March 2022 were included in the study.

**Data collection technique.** All patients who were diagnosed to have prosthetic stuck valve during the mentioned period were included in the study. Patient registration log book was used to identify patients who were treated for prosthetic stuck valve in the center. Then data were collected by reviewing their charts. Demographic characteristics (age and sex), and clinical profiles (underlying valvular lesion, type of mechanical valve implanted, anticoagulant adherence, International Normalized Ratio (INR), left ventricular function, presenting symptom, echo findings, thrombolytic treatment, patient outcome) were collected by reviewing the charts. Information that could help authors identify individual patients were not included in the data collection.

We defined *prosthetic valve* as mechanical or bio prosthetic valve implanted for patients who had native valve disease and who developed *prosthetic stuck valve* as a patient with acute onset of hemodynamic instability, echocardiographic findings with elevated trans-prosthetic valve pressure gradient, central regurgitation, immobility or reduced leaflet mobility, and the presence of thrombus on either side of the prosthesis, with or without obstruction [2]. Patients who were clinically asymptomatic while having immobile prosthetic valve disc and/or elevated trans-prosthetic gradient seen on echo were also included.

**Patient management protocol for prosthetic stuck valve at the Cardiac Center of Ethiopia.** Patient who is clinically suspected to have obstructive thrombotic prosthetic valve will undergo Transthoracic Echocardiography (TTE). If possible, Trans-esophageal Echocardiography (TEE) and cine fluoroscopy are also done. Once patient is diagnosed with stuck thrombotic valve, laboratory tests such as Complete Blood Count (CBC), coagulation profile, renal function test and chest X-ray will be done. In the meantime, patient will be admitted to Intensive Care Unit (ICU) for monitoring and informed consent will be sought. Patient will be assessed for an increased risk of bleeding (such as low platelet count, presence of bleeding diathesis, INR value above the target range) and if the International Normalized Ratio (INR) is found to be less than 2, in most cases streptokinase with loading dose of 250,000 IU over one hour and then maintenance dose of 100,000 IU/hour via infusion will be initiated.

TTE will be done every 6 hours to assess mobility and the mean gradient across the stuck valve. If the valve is opened within 24 hrs of streptokinase initiation, we stop the streptokinase and then start heparin infusion till the INR target for individual prosthetic valve is reached. If the stuck valve is not opened with in 24 hrs of streptokinase initiation, we continue the streptokinase infusion for 48–72 hrs. During the infusion, major adverse effect will be monitored. Redo valve surgery will be offered if there is no response to the thrombolytic therapy administered for 72 hrs. For some reasons, if thrombolytic agent is not available or partial obstruction of the valve with mild symptoms is diagnosed, then heparin infusion will be given with target partial PTT 1.5-2x of the upper normal for the patient. PTT will be determined every 6 hrs. At the end, cine fluoroscopy will be done. We use Tenecteplase when patient has previous exposure to streptokinase (in the past one year) in order to avoid hypersensitivity reaction to streptokinase re-administration. Additionally, tenecteplase is more expensive in our setting and not routinely available for first line use.

**Data analysis.** Data were analyzed by using Statistical Package for Social Sciences (SPSS) version 20.0 for windows. Mean and median values were determined for continuous variables. Data were presented using frequency tables and in text form.

**Ethics statement.** The study was approved by the Institutional Review Board (IRB) of St Paul's Hospital Millennium Medical College (ethical approval code = pm23/320). Informed consent was waived by the Institutional Review Board (IRB) of St Paul's Hospital Millennium Medical College due to the retrospective nature of the study. The study was conducted in accordance with the Ethiopian national research guideline and the Helsinki Declaration.

## Results

A total of 11 patients (13 episodes of stuck valve) were treated for prosthetic stuck valves between the periods July 2017 through March 2022. During the same period, a total of 133 patients underwent prosthetic valve replacement surgery making the prevalence of PVT to be 8.3%. The median age was 28 years old (IQR 22.5–34.0) and nine of them were female. The youngest and oldest patients were 18 and 46 years old respectively.

All patients had bi-leaflet prosthetic mechanical valves. The prosthetic valve was at the mitral valve position in ten patients, mitral and aortic valve positions in two patients, and at the aortic valve position in one patient. However, only one patient had the prosthetic stuck valve at the aortic valve position. The median duration of mechanical valve implantation before having a prosthetic stuck valve was 36 months (IQR 5–72) with the shortest and longest durations being 2 months and 108 months respectively.

Just before the onset of the prosthetic stuck valve, all patients had good adherence to anticoagulant medication (all were taking only warfarin); yet only five of them had their INR value in the recommended range. Nine patients presented with class III/IV New York Heart Association (NYHA) heart failure symptoms (median duration of symptom was three days) due to the prosthetic stuck valve and for the rest four patients, the stuck valve was detected incidentally and the patients had class I/II NYHA symptoms when they came for the regular follow-up.

Pre-intervention/pre-treatment echocardiographic study showed reduced mobility of at least one of the discs of the prosthetic mechanical valve in all of the patients. However, only six patients underwent pre-treatment Fluoroscopy study and in all of them, it was seen that at least one of the discs of the prosthetic valve was immobile. The remaining seven patients didn't undergo fluoroscopy study due to their hemodynamic instability and occasionally due to malfunction of the fluoroscopy machine.

Eleven patients received thrombolytic therapy alone as first-line treatment and nine of them responded to it. One of the patient who failed to respond to thrombolytic therapy underwent surgical mechanical valve replacement therapy. Of the 13 patients, two of them responded to combination therapy of heparinization with optimization of anticoagulant therapy and didn't require thrombolytic therapy. Of the total eleven patients who received thrombolytic therapy, ten of them received streptokinase and the remaining one patient received Tenecteplase. Of the ten patients who received streptokinase, two of them developed fever and one patient developed bleeding (excessive epistaxis and gum bleeding) as a complication of the treatment (Table 1). All the patients survived hospital discharge. The median duration of hospital stay was eight days (IQR 5.25–11.0).

The mean value of the mean trans-mitral valve gradient of the patients who had isolated mitral prosthetic stuck valve (12 patients) before initiation of therapy for the stuck valve was 18.3mmHG and the post-treatment mean gradient for the group was 7.2mmHG.

Two of the patients (patient 4a & 4b and patient 6a & 6b) had recurrence of the prosthetic stuck valve. One of them developed a recurrence of the stuck valve three months after the initial episode and the other patient had it three years after the initial episode (Table 1).

## Discussion

The current study aimed to describe treatment outcome of patients who had mechanical prosthetic stuck valve. Most of the patients were female and most stuck valves happened at the mitral valve position. We also found out that the INR values of most of the patients were below the recommended target value. Most patients responded to medical treatment alone and there was no death among the case series we studied.

**Table 1. Clinical and echocardiographic characteristics of patients with mechanical stuck valve, 2022.**

| Patient | Characteristics | | | | | | | | |
|---|---|---|---|---|---|---|---|---|---|
| | Age | Sex | Duration since valve implanted | INR[#] | Site of mechanical valve | Site of stuck valve | Mean gradient (before)* | Mean gradient (after)** | Treatment |
| 1 | 18 | F | 56month | 1.3 | Mitral | Mitral | 15 | 4.2 | Thrombolysis |
| 2 | 37 | F | 2month | 1.5 | Mitral | Mitral | 32 | 6.0 | Thrombolysis |
| 3 | 20 | M | 6month | 1.5 | Mitral | Mitral | 12 | 2.8 | Heparin & anti-coagulant |
| [#]4a | 31 | F | 60month | 1.8 | Mitral | Mitral | 17 | 6.0 | Surgery |
| [#]4b | 31 | F | 3month | 2.9 | Mitral & aortic | Mitral | 31 | 9.0 | Thrombolysis |
| 5 | 37 | F | 84month | 3.9 | Mitral & aortic | Mitral | 30 | 4.3 | Thrombolysis |
| [$]6a | 26 | F | 26month | 1.7 | Mitral | Mitral | 12 | 7.6 | Thrombolysis |
| [$]6b | 29 | F | 51month | 1.6 | Mitral | Mitral | 30 | 20.0 | Thrombolysis |
| 7 | 28 | M | 36month | 3.2 | Mitral | Mitral | 16 | 4.4 | Thrombolysis |
| 8 | 46 | F | 10month | 3.1 | Mitral | Mitral | 7 | 4.5 | Heparin & anti-coagulant |
| 9 | 25 | F | 4month | 2.4 | Mitral | Mitral | 15 | 7.0 | Thrombolysis |
| 10 | 27 | M | 84month | 2.1 | Aortic | Aortic | 60 | 29.0 | Thrombolysis |
| 11 | 18 | M | 108month | 1.74 | Mitral | Mitral | 30 | 10.8 | Thrombolysis |

[#]INR- International Normalized Ratio; Mean trans-vulvar gradient *before intervention and **after intervention.

[#]4a and [#]4b as well as [$]6a and [$]6b represent patients who had two episodes of stuck valve.

In line with the findings of the current study, previous study reported female patients with prosthetic mechanical valve to have higher likelihood of developing PVT as compared to their male counter parts [11].

Due to hemodynamic characteristic of the prosthetic valve, mitral prosthetic valve thrombosis is 2–3 times more frequent than thrombosis of aortic valve prosthesis [2]. That explains the higher number of patients with mitral valve prosthetic thrombosis than aortic valve prosthesis in our study population. Higher proportion of mitral valve prosthetic thrombosis than aortic valve prosthetic valve thrombosis was reported by a study conducted in India [12].

Patients with prosthetic mechanical valve require life-long anticoagulant therapy and it should be able to attain optimum INR level that is indicated for the site of the prosthetic mechanical valve [13]. However, patients in resource limited settings lack proper lab monitoring of anticoagulant therapy due to different social and economic reasons. In our case series, most patients had suboptimal anticoagulant therapy as reflected by their low INR level though they claim to have good adherence to the treatment regimen. We believe that the suboptimal anticoagulant therapy is contributory to the development of PVT in our study population.

In the current study, the median time between valve replacement and onset of PVT was 36 months. Previous study reported median time of four years. The risk of PVT is high in the first year of the postoperative period (24%) with stable incidence between the second to fourth years (15%) and declining incidence then after [14]. In our case series, the occurrence of PVT can be related to the inadequate dose of anti-coagulation therapy they were receiving at the time of their presentation.

In our case series, the fact that most of the patients responded to medical therapy alone is interesting finding. For left sided obstructive PVT, surgery is the option of treatment and fibrinolysis is reserved for patients with poor functional class and those who are high risk to undergo surgery [15]. Previous study indicated that the success rate of fibrinolysis therapy alone is 82% (2,6). The higher rate of response to medical treatment of stuck valve management

in our study population is an interesting finding since cardiac surgery is expensive and unavailable for most patients in resource limited settings such as ours. As a result, in low income countries, it is estimated that only about 11.0% of cardiac patients who demand cardiac surgery are operated on [16].

Limitation of the study includes small size study population and single institution study which makes the findings difficult to be generalized.

## Conclusion

Prosthetic valve thrombosis can happen several months after valve replacement surgery is done and it is accompanied by sub-optimal anticoagulant therapy. Most patients responded to medical therapy alone.

## Supporting information

**S1 Appendix. Anonymized data set used for the current study.**
(SAV)

## Author Contributions

**Conceptualization:** Kefelegn Dejene Tadesse, Meron Gebru, Atnafu Mekonnen Tekleab.

**Data curation:** Kefelegn Dejene Tadesse, Meron Gebru, Atnafu Mekonnen Tekleab.

**Formal analysis:** Atnafu Mekonnen Tekleab.

**Investigation:** Kefelegn Dejene Tadesse, Meron Gebru, Atnafu Mekonnen Tekleab.

**Methodology:** Kefelegn Dejene Tadesse, Meron Gebru, Atnafu Mekonnen Tekleab.

**Resources:** Kefelegn Dejene Tadesse, Meron Gebru.

**Supervision:** Kefelegn Dejene Tadesse.

**Validation:** Kefelegn Dejene Tadesse, Meron Gebru, Atnafu Mekonnen Tekleab.

**Visualization:** Meron Gebru, Atnafu Mekonnen Tekleab.

**Writing – original draft:** Kefelegn Dejene Tadesse, Meron Gebru, Atnafu Mekonnen Tekleab.

**Writing – review & editing:** Kefelegn Dejene Tadesse, Meron Gebru, Atnafu Mekonnen Tekleab.

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
