## [Decision Letter · Decision Letter 0]

31 Jan 2023

PONE-D-22-34873Treatment outcome of patients with prosthetic stuck valves at the Cardiac Center of EthiopiaPLOS ONE

Dear Dr. Tekleab,

Thank you for submitting your manuscript to PLOS ONE. After careful consideration, we feel that it has merit but does not fully meet PLOS ONE’s publication criteria as it currently stands. Therefore, we invite you to submit a revised version of the manuscript that addresses the points raised during the review process.

We look forward to receiving your revised manuscript.

Kind regards,

Justin Paul Gnanaraj, MD, DM

Academic Editor

PLOS ONE

Journal Requirements:

"I have read the journal's policy and one of the authors of this manuscript have the following competing interest: Dr Atnafu Mekonnen Tekleab is serving as Academic Editor of PLOS ONE Journal."

Additional Editor Comments:

In addition to responding to the queries of the reviewers, the authors are requested to highlight if there is any new contribution, this study offers or if there is any specific regional perspective this study brings about.

Reviewers' comments:

Reviewer's Responses to Questions

**Comments to the Author**

1. Is the manuscript technically sound, and do the data support the conclusions?

Reviewer #1: Partly

Reviewer #2: Partly

2. Has the statistical analysis been performed appropriately and rigorously? 

Reviewer #1: N/A

Reviewer #2: I Don't Know

3. Have the authors made all data underlying the findings in their manuscript fully available?

Reviewer #1: Yes

Reviewer #2: Yes

4. Is the manuscript presented in an intelligible fashion and written in standard English?

Reviewer #1: Yes

Reviewer #2: Yes

5. Review Comments to the Author

Reviewer #1: The sample volume is small and single Center as stated by the authors themselves.

The total no of patients , who had valve surgery during the study period can be provided, to calculate the incidence and prevalence of prosthetic valve thrombosis

The clinical characteristics and pattern of anticoagulation and the INR and TTR status of all the patients who had valve replacements during this period, will be of great value in statistical analysis and to identify the predictors of PVT

Reviewer #2: Introduction:

P3 Line 6-7: The author mentions surgical approach, fibrinolysis, and heparin treatment are considered as treatments of choice. As per guidelines, for patients with a thrombosed left-sided mechanical prosthetic heart valve who present with symptoms of valve obstruction, urgent initial treatment with either slow-infusion, low- dose fibrinolytic therapy or emergency surgery is recommended. Heparinisation is not a treatment of choice.

Methods:

P3 Line 27-29: The following phrase need not be mentioned in study setting “ Following the valve replacement surgery, patients might develop stuck valve and present with an emergency condition”

Study design and study period:

P4 Line 2: The actual study design should be detailed.

Data collection technique:

P4 Line 6: The inclusion criteria not properly delineated- whether all the consecutive patients with stuck valve during the study period were included. Any exclusion criteria to be added. “Information that could help authors identify individual patients were not included in the data collection”- requires clarity

P4 Line 8-12: The author mentions the data collected in the given lines.

Has the data on immediate post operative trans prosthetic valvular gradient or the gradients recorded prior to the PVT during the previous routine visits collected?

Any pre study period complications (bleeding or thrombotic) available for patients who had prosthetic valve implanted prior to the study period?

P4 Line 10: The data on anticoagulant adherence is collected but the type of anticoagulant and the dosage, the patient is on, has not been mentioned? Were all the patient on warfarin?

P4 Line 11: Instead of thrombolytic treatment, interventions done can be mentioned since not all the patients were thrombolysed.

P4 Line 15-18: Was the study entirely based on clinical and echocardiographic criteria? Was fluoroscopic criteria included in the study?

What does the author mean by acute hemodynamic instability as the clinical criteria- the author also mentions “four patients, the stuck valve was detected incidentally when they came for the regular follow-up” (P6 Line 6-7). Kindly clarify the definition for stuck valve.

The patient symptoms on the basis of NHYA is not mentioned.

The echocardiographic criteria mentions a series of findings- where they all pre requisite for diagnosis of stuck valve in this study or a combination of the findings were alone necessary- clarification needed.

Patient management protocol for prosthetic stuck valve:

P4 Line 20: The author mentions that this is the protocol for patients with obstructive thrombotic prosthetic valve. In the previous section while defining stuck prosthetic valve it is mentioned with or without obstruction(P4 Line 18). Is there different protocol for patients without obstruction? What is their symptom profile?

P4 Line 27: Patient will be assessed for an increased risk of bleeding- Kindly mention what criteria/ scoring was used to assess the bleeding risk.

P4 Line 28-29: Mentions only the protocol for thrombolysis with streptokinase. “Ten of them received streptokinase and the remaining one patient received Tenecteplase” What was the protocol used for TNK managed patient?

P5 Line 6: “Redo valve surgery will be offered if there is no response to the thrombolytic therapy” Is this after 72 hrs of thrombolytic therapy. Kindly clarify

P5 Line 9: “At the end, cine fluoroscopy will be done”. Is fluoroscopy done in all cases?

Results:

P5 Line 27-28: “However, only one patient had the prosthetic stuck valve at the aortic valve position” The sample size appear to be too small to comment on this.

P6 Line 4-7: “Nine patients presented with symptoms of heart failure (median duration was three days) due to the prosthetic stuck valve and for the rest four patients, the stuck valve was detected incidentally when they came for the regular follow-up”

Does this mean that the incidentally diagnosed patients were also in acute hemodynamic compromise as mentioned in the definition of stuck prosthetic valve?

P6 Line 9: “However, only six patients underwent pre-treatment Fluoroscopy study”. Was hemodynamic instability the reason for fluoroscopy not done in other patient- reasons can be mentioned.

P6 Line 17-19

“Of the ten patients who received streptokinase, two of them developed fever and one patient developed bleeding as a complication of the treatment”

What type of bleeding complication was encountered? Was it a major or a minor bleed and how was it managed

P6 Line 21-23: “The mean value of the mean trans-mitral valve gradient of the patients who had isolated mitral prosthetic stuck valve (12 patients) before therapy was initiated was 18.3mmHG and the post- treatment mean gradient for the group was 7.2mmHG”.

Was the post operative gradients available for these patients. For patients for whom baseline value was not available, what was the criteria used to confirm prosthetic valve obstruction?

P6 Line 24-26:

“Two of the patients (patient 4 & 5 and patient 7 & 8) had recurrence of the prosthetic stuck valve. One of them developed a recurrence of the stuck valve three months after the initial episode and the other patient had it three years after the initial episode”

Does this mean 11 patients and 13 episodes- kindly clarify

6. PLOS authors have the option to publish the peer review history of their article (what does this mean?). If published, this will include your full peer review and any attached files.

Reviewer #1: **Yes: **T R Muralidharan

Reviewer #2: No

---

## [Author Response · Author response to Decision Letter 0]

12 Mar 2023

Dear Editor and Reviewers,

Thank you so much. We have provided a Point_by_point response to your comments. We have uploaded all the necessary information and data.

---

## [Decision Letter · Decision Letter 1]

5 Apr 2023

Treatment outcome of patients with prosthetic stuck valves at the Cardiac Center of Ethiopia

PONE-D-22-34873R1

Dear Dr.Atnafu Mekonnen Tekleab,

We’re pleased to inform you that your manuscript has been judged scientifically suitable for publication and will be formally accepted for publication once it meets all outstanding technical requirements.

Kind regards,

Justin Paul Gnanaraj, MD, DM

Academic Editor

PLOS ONE

Reviewers' comments:

Reviewer's Responses to Questions

**Comments to the Author**

1. If the authors have adequately addressed your comments raised in a previous round of review and you feel that this manuscript is now acceptable for publication, you may indicate that here to bypass the “Comments to the Author” section, enter your conflict of interest statement in the “Confidential to Editor” section, and submit your "Accept" recommendation.

Reviewer #1: All comments have been addressed

Reviewer #2: All comments have been addressed

2. Is the manuscript technically sound, and do the data support the conclusions?

Reviewer #1: Partly

Reviewer #2: Partly

3. Has the statistical analysis been performed appropriately and rigorously? 

Reviewer #1: N/A

Reviewer #2: Yes

4. Have the authors made all data underlying the findings in their manuscript fully available?

Reviewer #1: Yes

Reviewer #2: Yes

5. Is the manuscript presented in an intelligible fashion and written in standard English?

Reviewer #1: Yes

Reviewer #2: Yes

6. Review Comments to the Author

Reviewer #1: (No Response)

Reviewer #2: (No Response)

7. PLOS authors have the option to publish the peer review history of their article (what does this mean?). If published, this will include your full peer review and any attached files.

Reviewer #1: **Yes: **Thoddi Ramamurthy Muralidharan

Reviewer #2: No

---

## [Editor Report · Acceptance letter]

12 Apr 2023

PONE-D-22-34873R1 

Treatment outcome of patients with prosthetic stuck valves at the Cardiac Center of Ethiopia 

Dear Dr. Tekleab:

I'm pleased to inform you that your manuscript has been deemed suitable for publication in PLOS ONE. Congratulations! Your manuscript is now with our production department. 

Kind regards, 

on behalf of

Dr. Justin Paul Gnanaraj 

Academic Editor

PLOS ONE